



**Long-term Land-Atmosphere Energy and Water Exchange Observational Dataset over central Tibetan Plateau**

Haipeng Yu[1], Guantian Wang[1,4], Zeyong Hu[1]*, Yaoming Ma[2], Maoshan Li[3], Weiqiang Ma[2], Lianglei
Gu[1], Fanglin Sun[1], Hongchun Gao[1], Shujin Wang[1], Fuquan Lu[1,4]
[1] Nagqu Plateau Climate and Environment Observation and Research Station of Tibet Autonomous
Region, State Key Laboratory of Cryospheric Science and Frozen Soil Engineering, Northwest Institute
of Eco-Environment and Resources, Chinese Academy of Sciences, Lanzhou 730000, China
[2]Land-Atmosphere Interaction and Its Climatic Effects Group, State Key Laboratory of Tibetan Plateau
Earth System, Environment and Resources, National Observation and Research Station for
Qomolongma Special Atmospheric Processes and Environmental Changes, Institute of Tibetan Plateau
Research, Chinese Academy of Sciences, Beijing 100101, China
[3]School of Atmospheric Sciences, Plateau Atmosphere and Environment Key Laboratory of Sichuan
Province, Chengdu Plain Urban Meteorology and Environment Observation and Research Station of
Sichuan Province, Chengdu University of Information Technology, Chengdu 610225, China
[4]University of Chinese Academy of Sciences, Beijing 100049, China
*Correspondence to*:  Zeyong Hu (zyhu@lzb.ac.cn)
**Abstract**:In recent decades, the climate of Tibetan Plateau has undergone notable changes, which has
a strong influence on global climate systems and human activities, making it a research hotspot.
However, due to its extreme elevation, harsh environment, and complex underlying surface, long-term
observations of the central Plateau's atmospheric vertical profiles have been challenging and scientific
data sharing is crucial and in urgent need. This paper presents a 9-year observational dataset (2014-
2022) with hourly temporal resolution from the Nagqu region of northern Tibet. The dataset is a
combination of four field stations covering the central Tibetan Plateau, which recorded near-surface
meteorological data, radiation budget, turbulent fluxes, and soil hydrothermal characteristics. All
observational items in this dataset underwent data processing and quality control to ensure data quality.
This dataset represents the most detailed raw observational data on the central Tibetan Plateau's spatial
coverage and recent changes. It holds significant value in revealing energy and water exchanges
between the land surface and the atmosphere on the Tibetan Plateau. Main datasets are freely available
at     the     National     Tibetan     Plateau     /     Third     Pole     Environment     Data     Center
(https://doi.org/10.11888/Meteoro.tpdc.270010, Hu et al., 2019) and additionally at National Tibetan
Plateau / Third Pole Environment Data Center (https://doi.org/10.11888/Atmos.tpdc.300325, Wang et
al., 2023).

**Key words:** field observation, observational dataset, land-atmosphere interaction, Tibetan Plateau,
energy and water exchanges

**1.Introduction**

The Tibetan Plateau (TP), with an average altitude exceeding 4000 meters, is the world's largest



mountain range, often referred to as the "Third Pole" (Qiu, 2008). It is the source of many major rivers,
including the Yangtze, Yellow, Mekong, and Brahmaputra, which provide freshwater to 1.4 billion
people (Immerzeel et al., 2020). It is also known as the "Asian Water Tower" (Immerzeel et al., 2010).
The plateau's substantial absorption of solar radiation renders it the warmest region in the mid-
troposphere. This plays a key role in the evolution of the East Asian Summer Monsoon and Indian
Summer Monsoon systems (Wu et al., 2007; Wu et al., 2012; Li et al., 2018; Zhang et al., 2019), which
in turn affect circulation patterns in East Asia, and even the Northern Hemisphere (Fan et al., 2023;
Curio and Scherer, 2016; Sun et al., 2022). The intricate surface characteristics of the TP make it
particularly susceptible to external disturbances in the context of global warming (Li et al., 2022; Kang
et al., 2010). Meteorological observations indicate that surface warming on the TP commenced at an
earlier and more accelerated pace than in other regions at the same latitude (Liu and Chen, 2000).
Climate change on the TP has resulted in most regions becoming warmer and moister, while the
southern part has become warmer and drier (Yang et al., 2014). These changes in temperature and
moisture have further led to significant shifts in many environmental systems on the plateau, including
precipitation (Yang et al., 2011), glaciers (Yao et al., 2012), lakes (Lei et al., 2014), and vegetation
(Shen et al., 2015). Therefore, in-depth research on TP not only helps us better understand the impact
of global warming on its unique ecosystems and climatic features, but also is crucial for assessing its
contribution to regional and even global water cycles and energy balance. In recent decades, a
multitude of comprehensive observation stations have been established with the objective of
strengthening climate change research on the TP. These include the Third Pole Environment
Observation and Research Platform and the Integrated Three-Dimensional Observation Research
Platform for Third Pole Environment (Ma et al., 2018; Ma et al., 2023). The establishment of these
observation stations has resulted in a significant improvement in the availability of data on land-
atmosphere interactions on the plateau. The comprehensive data network has enhanced the quality of
land surface models and corrected satellite retrievals (Ma et al., 2023). However, due to factors such as
transportation and maintenance costs, most observations are concentrated in the eastern part of the TP,
with limited long-term observations of the plateau's interior, especially in the high-altitude regions of
northern Tibet (Shen and Xiong,2016; Duan and Xiao, 2015).

The central Tibetan Plateau (CTP) belongs to a typical plateau climate with an average altitude
exceeding 4500 metres. It is bordered by the Kunlun and Tanggula Mountains to the northwest and the
Nyenchen Tanglha Mountains to the south, covering an area of 446,000 square kilometres, which
represents one-fifth of the total area of Tibet. The CTP is known as the "Roof of the Roof of the
World," representing the highest and most typical plateau form on the TP. Due to its high altitude, the
substrate is dominated by cold-resistant vegetation, with an annual precipitation of 154.9 mm and
approximately 50 rainy days, characterizing a typical subarctic semi-arid monsoon climate (Li et al.,
2017). During clear weather, the downward shortwave radiation peaks at 1200 W/m², which is
significantly higher than inland observations (He et al., 2010). The period from May to September is
characterised by warm and moist conditions, with frequent rainfall, while October to April is cold and
dry with prevalent strong winds. During the summer months, the CTP serves as a thermal core area,
with surface heating being a key factor in making the plateau a heat source. The thermal effect of the
CTP influences not only the plateau itself but also weather and climate systems in East Asia, and even
the Northern Hemisphere (Zhao et al.,2019). Over the past 40 years, the warming trend on the CTP has
been significantly higher than in lower-altitude areas, with surface temperature warming rates and
precipitation amounts also higher than in eastern regions of the TP (Chen et al., 2003; Liu et al., 2009;





Yang et al., 2002). The number of consecutive dry days on the central TP has been steadily decreasing
in conjunction with accelerated warming and moistening (Ciwangdunzhu et al., 2018), while its lakes
are experiencing strong expansion or contraction (Zhang et al., 2019; Lei et al., 2013 Lei et al., 2014).
Some lakes serve as heat sources year-round and have an annual evaporation rate of 312.9mm (Guo et
al., 2014). Concurrently, glaciers on the CTP are rapidly melting, and permafrost areas are shrinking,
severely impacting the surrounding environment (Jin et al., 2009; Sun et al., 2020). A substantial
evidence base indicates that the CTP 's ecological environment is highly fragile, and the surface heat
exchange in this region is significant for climate change. However, there remains substantial
uncertainty regarding the impact mechanisms (Song et al., 2012; Yang et al., 2014). This can be
attributed to the harsh environment, frequent extreme weather such as strong winds and hail showers,
and faulty observation equipment. Furthermore, much of the region's intricate topography is unsuitable
for human habitation, resulting in a sparsely populated area, inadequate transportation infrastructure,
and delayed equipment maintenance. This has led to the availability of relatively limited long-term and
stable observational data for the CTP.
To fill these gaps, a series of mesoscale observation networks have been established in the vicinity
of the Nagqu Observation Network of Plateau Climate and Environment (NPCE) (Xu et al., 2013). The
NPCE commenced observations in 1997 and serves as a link between the East Asian Monsoon and the
Indian Monsoon, as well as a transition zone between the monsoon systems and westerly circulation. It
is a region of significant thermal activity on the plateau, and the central zone for China's Third
Atmospheric Science Experiment on the TP. This distinctive geographical position gives NPCE a
pivotal role in TP research. Following decades of observations, data from the Nagqu observing network
has become an important platform for research on land-atmosphere interactions in northern Tibet. It has
provided significant support for studies on boundary layer observations (Gu et al., 2022; Ueno et al.,
2012), land-atmosphere energy transfer (Yang et al., 2010; Gu et al., 2015; Li et al., 2015), substrate
evaporation (Zou et al., 2018), soil hydrothermal balance (Bian et al., 2012; Fu et al., 2022), and
numerical simulations (Deng et al., 2021; Li et al., 2009; Sun et al., 2020). In recent years, there has
been a gradual increase in the availability of field datasets to the public with free access. In order to
optimise the potential value of scientific datasets, we describe and provide a high-time-resolution
dataset. This dataset comprises nine years of near-surface meteorological, radiation, turbulent flux, and
soil hydrothermal observations from NPCE. It provides a more accurate and comprehensive depiction
of land-atmosphere interactions and boundary layer structure on the TP, playing a key role in numerical
simulations and climate assessments.
The paper is organized as follows: Section 2 provides an overview of NPCE's geographic location,
observation equipment, variables, data processing, and data availability. Section 3 presents near-surface
meteorological, radiation, turbulent flux, and soil hydrothermal observations, discussing data integrity.
Section 4 comprises a discussion, and Section 5 is conclusion.

**2. Observation network and data processing**
**2.1 Site descriptions**
The NPCE is affiliated with the Northwest Institute of Eco-Environment and Resources, Chinese
Academy of Sciences, was established on 4 May 2009, with observational activities commencing on 1
April 1997. The central station is situated in Niaoqu Village, Luoma Town, Seni District, Nagqu City,
Tibet Autonomous Region (longitude 91.90° E, latitude 31.37° N, altitude 4509 metres), near the
Tanggula Mountains, where the Indian monsoon and the East Asian monsoon intersect. The



observation field is relatively flat and open, with abundant water resources and situated in an area
characterised by pastoral activities. The substrate vegetation is primarily alpine grassland, with a
relatively clean atmosphere and a low aerosol content. The station's unique geographic location and
subsurface characteristics are significant for the observation and study of high-altitude climate
environments.

Over more than 20 years, NPCE has developed into a network of four atmospheric boundary layer
physics and land surface process observation points in the Nagqu region and along the Qinghai-Tibet
Highway. The locations in Figure 1 are BJ/Nagqu, Amdo, NewD66/Kekexili, and MS3478/Liangdaohe.
These represent different substrate characteristics, namely plateau grassland, plateau meadow, plateau
wetland, and plateau bareland. In 2014, NPCE became a member of the Chinese Academy of Sciences
Cold Region Surface Process and Environmental Monitoring Research Network and the Chinese
Academy of Sciences Land Surface Process Research Network.

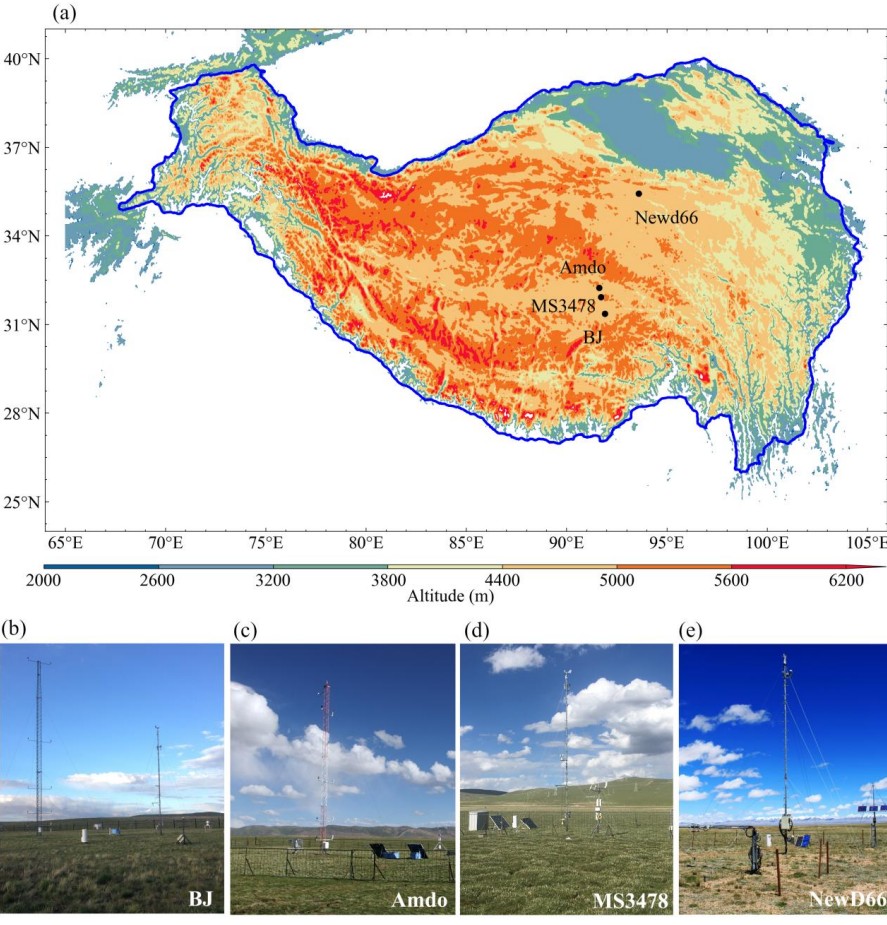

Figure 1. Geographic locations of the observation stations of NPCE (a) and on-site image of BJ (b),
Amdo (c), MS3478 (d), and NewD66 (e).



## 2.2 Observation infrastructure and data processing


The NPCE utilises automatic observation mode, capable of sustaining continuous observation
throughout the year. In order to ensure the continuity and accuracy of the observation data, the
maintenance engineers are responsible for conducting on-site inspections and maintenance of the
infrastructures at regular intervals at each observation station and collect the observation data. All data
are transmitted to the data centre for processing to facilitate timely and effective detection of technical
problems.
The NPCE near-surface meteorological observation system consists of Planetary Boundary Layer
(PBL) towers and Automatic Weather Stations (AWS). These instruments measure vertical profiles of
meteorological conditions, including air temperature, relative humidity, wind direction and wind speed
within the boundary layer. Observation heights range from 0.75-12 m at BJ and Amdo, and
approximately 1 m and 10 m at NewD66 and MS3478. The observation probes deployed at each station
were of the same type to ensure consistency, but the specific observation heights varied slightly across
different stations. The vertical soil hydrothermal characteristics were measured at each station with
multilayer soil temperature and moisture probes located between 5-160 cm from the surface, with a
proportional distribution of observation depths. High-frequency Eddy Covariance (EC) systems were
installed at the four stations after 2012, allowing for the capture of sensible, latent and carbon dioxide
fluxes, with a sampling frequency of 0.1 s and an output interval of 30 minutes. Surface observations
encompass surface air temperature, humidity, radiative quadrature, atmospheric pressure, precipitation,
and snow depth. All observations are recorded at a sampling frequency of 10 seconds and an output
interval of 30 minutes. More details of the observation instruments are listed in Table 1. Station BJ is
also equipped with a ground-based microwave radiometer, a blowing snow observation instrument, and
a wind profile radar. The ground-based microwave radiometer (MWP967KV) can observe temperature
and humidity profiles up to 10 km above the surface under all weather conditions, with a 2-minute
output interval. The FlowCapt FC4 blowing snow sensor inverts digital signals by collecting the
acoustic pressure of the blowing snow particles as they impact. The CFL-03 pulsed Doppler wind
profile radar (WPR) can capture the wind speed and wind direction profile. It has a detection range of
300 to 3,000 metres and an output interval of 5 minutes. These observational instruments collectively
constitute a well-established monitoring system capable of comprehensively tracking weather events
across the central TP and their associated impacts.
For near-surface micrometeorological observations in complex environments, quality control of
data processing is crucial. In order to ensure the consistency of data at each site in the harsh natural
environment of the TP, engineers firstly check the data in terms of range, which is a constant value set
according to the climatic conditions of the TP and ignore the seasonal variations. Factors such as the
replacement of original parts and extreme weather can lead to temporary sensor malfunction. This can
result in no change or unrealistic jumps in the data for several hours. These factors can make it difficult
to properly deal with the issue during the range check. Consequently, engineers will utilise manual
quality control at the conclusion of the process, based on empirical observations, to rectify or eliminate
anomalous intervals and thereby ensure that outliers do not adversely affect the mean observations. The
removed missing measurements and outliers are denoted by NAN or -9999. Moreover, three categories
of quality classification are employed for turbulent fluxes. 0 denotes the highest quality of flux, 1
denotes flux suitable for general analysis, and 2 denotes flux with possible anomalies.

Table 1. Detailed information about observation stations and instruments of NPCE.

| Site | Variables | Sensor models | Manufacturers | Observation | Installation height/depth (m) | Units |
|------|-----------|---------------|---------------|-------------|-------------------------------|-------|





| | | | | period | | |
|---|---|---|---|---|---|---|
| **BJ**<br>Lat: 31.37°N<br>Lon: 91.90°E<br>Altitude: 4509 m<br>Established in 2001<br>Plateau Grassland | Air temperature | HMP45D | Vaisala | 2001-2018 | 1.0/2.0/4.0/8.0/16.0/24.0 | °C |
| | | HMP155 | Vaisala | 2012-2022 | 0.75/1.5/3.0/6.0/12.0/22.0 | |
| | Wind speed and direction | A-5400 | Komatsu | 2001-2018 | 10.36 | m/s |
| | | WindSonic | Gill | 2012- 2022 | 0.75/1.5/3.0/6.0/12.0/22.0 | |
| | Relative humidity | HMP45D | Vaisala | 2001-2018 | 1.0/2.0/4.0/8.0/16.0/24.0 | % |
| | | HMP155 | Vaisala | 2012-2022 | 0.75/1.5/3.0/6.0/12.0/22.0 | |
| | Precipitation | NOAH-II | ETI | 2001-2018 | 1.00 | mm |
| | | T-200B | Geonor | 2012-2022 | 1.00 | |
| | Soil temperature | TS-301 | Okazaki | 2001-2018 | -0.04/-0.10/-0.20/-0.40 | °C |
| | | TR-219 | Truwel | 2012-2022 | -0.05/-0.10/-0.20/-0.40/-0.80/-1.60 | |
| | Soil volumetric water content | Trime EZ | IMKO | 2001-2018 | -0.04/-0.20 | m³/m³ |
| | | CS616 | Campbell | 2012-2022 | -0.05/-0.10/-0.20/-0.40/-0.80/-1.60 | |
| | Pressure | PTB220C | Vaisala | 2001-2022 | 0.50 | hPa |
| | Downward shortwave radiation | CM21 | Kipp&Zonen | 2001-2022 | 1.50 | W/m² |
| | Upward shortwave radiation | CM21 | Kipp&Zonen | 2001-2022 | 1.50 | W/m² |
| | Downward longwave radiation | PIR | Eppley | 2001-2022 | 1.50 | W/m² |
| | Upward longwave radiation | PIR | Eppley | 2001-2022 | 1.50 | W/m² |
| | Surface sensible heat flux | CSAT3 | CAMPBELL | 2008-2022 | 3.00 | W/m² |
| | Surface latent heat flux | LI7500A<br>HMP45AC | CAMPBELL<br>VAISALA | | | |
| **Amdo**<br>Lat: 32.24°N<br>Lon: 91.62°E<br>Altitude: 4695 m<br>Established in 1997<br>Plateau Meadow | Air temperature | HMP155 | Vaisala | 2012-2022 | 1.5/3.0/6.0/12.0 | °C |
| | Wind speed and direction | WindSonic | Gill | 2012-2022 | 1.5/3.0/6.0/12.0 | m/s |
| | Relative humidity | HMP155 | Vaisala | 2012-2022 | 1.5/3.0/6.0/12.0 | % |
| | Precipitation | T-200B | Geonor | 2012-2022 | 1.00 | mm |
| | Soil temperature | TR-219 | Truwel | 2012-2022 | -0.05/-0.10/-0.20/-0.40/-0.80/-1.60 | °C |
| | Soil volumetric water content | CS616 | Campbell | 2012-2022 | -0.05/-0.10/-0.20/-0.40/-0.80/-1.60 | m³/m³ |
| | Pressure | PTB220C | Vaisala | 2001-2022 | 0.50 | hPa |
| | Downward shortwave radiation | MS-801 | EKO | 2012-2022 | 1.50 | W/m² |
| | Upward shortwave radiation | MS-801 | EKO | 2012-2022 | 1.50 | W/m² |
| | Downward longwave radiation | PIR | EPPLEY | 2012-2022 | 1.50 | W/m² |
| | Upward longwave radiation | PIR | EPPLEY | 2012-2022 | 1.50 | W/m² |
| | Surface sensible heat flux | CSAT3 | CAMPBELL | 2012-2022 | 3.00 | W/m² |
| | Surface latent heat flux | LI7500A<br>HMP45AC | CAMPBELL<br>VAISALA | | | |
| **MS3478**<br>Lat: 31.93°N<br>Lon: 91.71°E<br>Altitude: 4620 m<br>Established in 2001<br>Plateau Wetland | Air temperature | HMP45D | Vaisala | 2001-2022 | 0.85/9.35 | °C |
| | Wind speed and direction | A-5400 | Komatsu | 2001-2022 | 10.35 | m/s |
| | Relative humidity | HMP45D | Vaisala | 2001-2022 | 0.85/9.35 | % |
| | Precipitation | NOAH-II | ETI | 2001-2022 | 1.00 | mm |
| | Soil temperature | TS-301 | Okazaki | 2001-2022 | -0.04/-0.10/-0.20/-0.40 | °C |
| | Soil volumetric water content | Trime EZ | IMKO | 2001-2022 | -0.04/-0.20 | m³/m³ |
| | Pressure | PTB220C | Vaisala | 2001-2022 | 0.5 | hPa |
| | Downward shortwave radiation | CM21 | Kipp&Zonen | 2001-2022 | 1.50 | W/m² |
| | Upward shortwave radiation | CM21 | Kipp&Zonen | 2001-2022 | 1.50 | W/m² |
| | Downward longwave radiation | PIR | Eppley | 2001-2022 | 1.50 | W/m² |
| | Upward longwave radiation | PIR | Eppley | 2001-2022 | 1.50 | W/m² |
| | Surface sensible heat flux | CSAT3 | CAMPBELL | 2012-2022 | 3.00 | W/m² |
| | Surface latent heat flux | LI7500A<br>HMP45AC | CAMPBELL<br>VAISALA | | | |
| **NewD66**<br>Lat: 35.43°N<br>Lon: 93.59°E<br>Altitude: 4465 m<br>Established in 1997<br>Relocate in 2006<br>Plateau Bareland | Air temperature | HMP45D | Vaisala | 1997-2022 | 0.60/9.30 | °C |
| | Wind speed and direction | W955-FE | Komatsu | 1997-2022 | 10.40 | m/s |
| | Relative humidity | HMP45D | Vaisala | 1997-2022 | 0.60/9.30 | % |
| | Precipitation | NOAH-II | ETI | 1997-2022 | 1.00 | mm |
| | Soil temperature | TS-301 | Okazaki | 1997-2022 | 0.00/-0.01/-0.02/-0.03/-0.04/-0.06/-0.08/-0.10/-0.15/-0.20/-0.40/-0.60/-0.80/-1.00/-1.30/-1.60/-2.00/-2.12 | °C |
| | Soil volumetric water content | Trime EZ | IMKO | 1997-2022 | -0.011/-0.04/-0.075/-0.20/-0.40/-0.60/-0.80/-1.00/-1.60/-2.06 | m³/m³ |
| | Pressure | PTB210 | Vaisala | 1997-2022 | 0.5 | hPa |
| | Downward shortwave radiation | CM21 | Kipp&Zonne | 1997-2022 | 1.50 | W/m² |
| | Upward shortwave radiation | CM21 | Kipp&Zonne | 1997-2022 | 1.50 | W/m² |
| | Downward longwave radiation | PIR | Eppley | 1997-2022 | 1.50 | W/m² |
| | Upward longwave radiation | PIR | Eppley | 1997-2022 | 1.50 | W/m² |
| | Surface sensible heat flux | CSAT3 | CAMPBELL | 2012-2022 | 3.00 | W/m² |
| | Surface latent heat flux | LI7500A<br>HMP45AC | CAMPBELL<br>VAISALA | | | |



**3. Data description**
**3.1 Near-surface meteorological data (Air temperature, relative humidity, wind speed)**

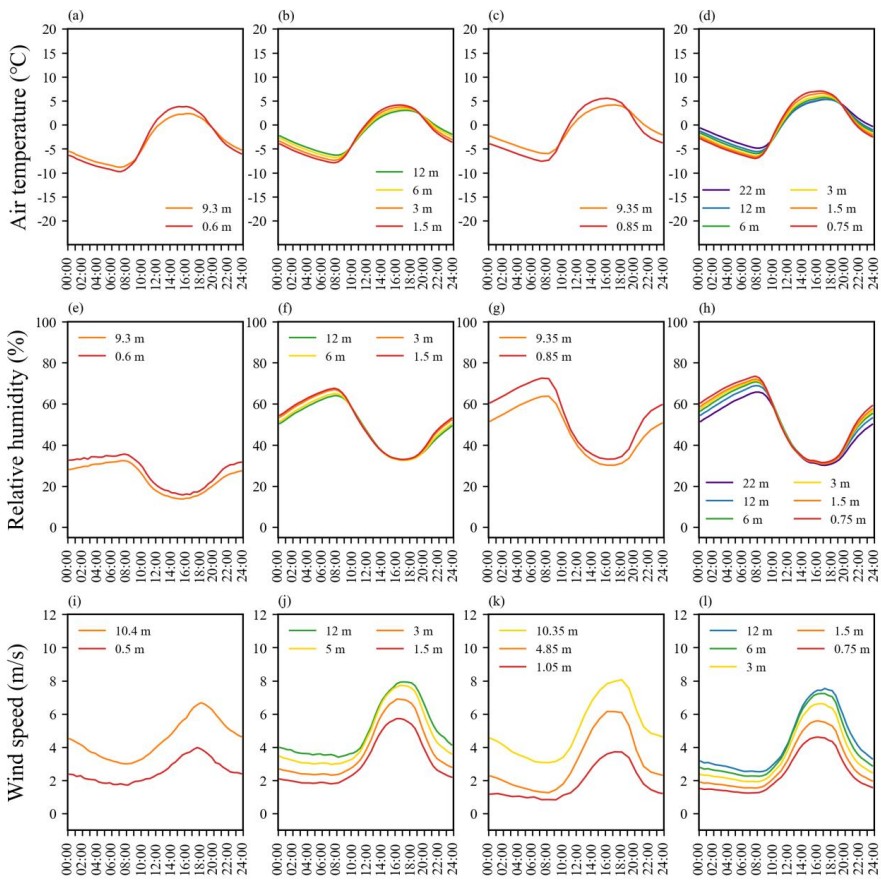

Figure 2. Diurnal variations of temperature at different heights of the PBL tower at NewD66 (a), Amdo (b), MS3478 (c) and BJ (d), with corresponding relative humidity (efgh) and wind speed (ijkl) at each station.

The main variables measured in near-surface micrometeorology are surface temperature, relative humidity and wind speed. Surface air pressure is a single-level observation, which decreases with altitude, while the other variables are gradient observations. Fig. 2 compares the diurnal cycles of temperature, relative humidity and wind speed at different sites to illustrate the micrometeorological characteristics influenced by the different underlying surfaces. Most sites exhibit a maximum temperature at 18:00 and a minimum at 09:00, indicating a clear diurnal pattern. As a result of latitudinal effects, the average temperature at the stations increases from north to south. The northernmost station NewD66 records a minimum temperature of -9.73 °C and a maximum of 3.84 °C, while the southernmost station BJ records a minimum temperature of -7.02 °C and a maximum of 7.05 °C. Irrespective of the water vapour content, relative humidity tends to decrease during the day and increase at night. The most pronounced diurnal variations are observed at lower altitudes. This is because daytime solar radiation heats the surface and atmosphere, making it harder for lower air layers to saturate, which naturally reduces relative humidity. The wind speed generally increases with altitude

and shows a diurnal variation, being lower at night and higher during the day. MS3478 is situated in a
plateau wetland, where the abundance of vegetation increases friction with the underlying surface,
resulting in significantly reduced surface wind speeds at this station in comparison to other
locations.

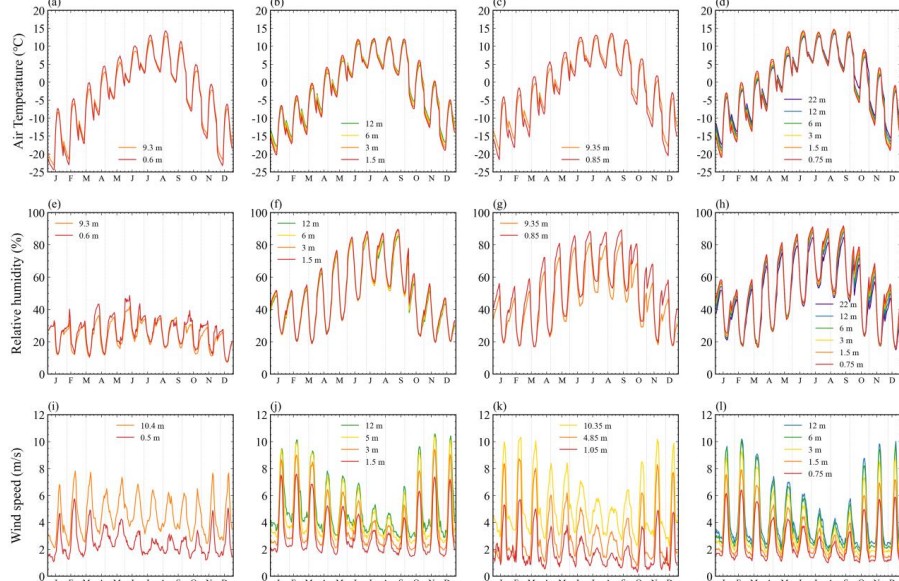

Figure 3. Monthly diurnal variations of temperature at different heights of the PBL tower at NewD66
(a), Amdo (b), MS3478 (c) and BJ (d), with corresponding relative humidity (efgh) and wind speed
(ijkl) at each station.

Most stations exhibited notable seasonal fluctuations in temperature, relative humidity and wind
speed (Fig. 3). Due to its cold bareland substrate, the NewD66 station experiences more pronounced
diurnal temperature variations, with nighttime temperatures lower than those observed at other stations,
occasionally reaching below -20 ℃ in January. In summer, with less pronounced diurnal variations
than in winter, temperatures peak at all sites, reaching approximately 13 ℃ during the day and
approximately 6 ℃ at night. Diurnal variations in relative humidity also show seasonal patterns, with
NewD66 remaining consistently lower, never exceeding 50%, and showing small variations in spring
and even smaller variations in summer. In contrast, the most pronounced diurnal variations in relative
humidity are observed during the spring at the other stations. As water vapour increases over the
plateau, nighttime relative humidity remains above 50% at all sites in summer, with diurnal humidity
variations gradually decreasing. Wind speeds show greater diurnal variation in winter and increase with
altitude. Peak wind speeds of more than 10 m/s are observed at the Amdo and BJ stations. In summer,
the 12 m wind speed at the Amdo and BJ sites peaks at 4.31 m/s in August. Surface wind speeds at
MS3478 are consistently low, but upper-level wind speeds are similar to those observed at the other
sites. Consequently, MS3478 exhibits the most pronounced gradient in wind speed.



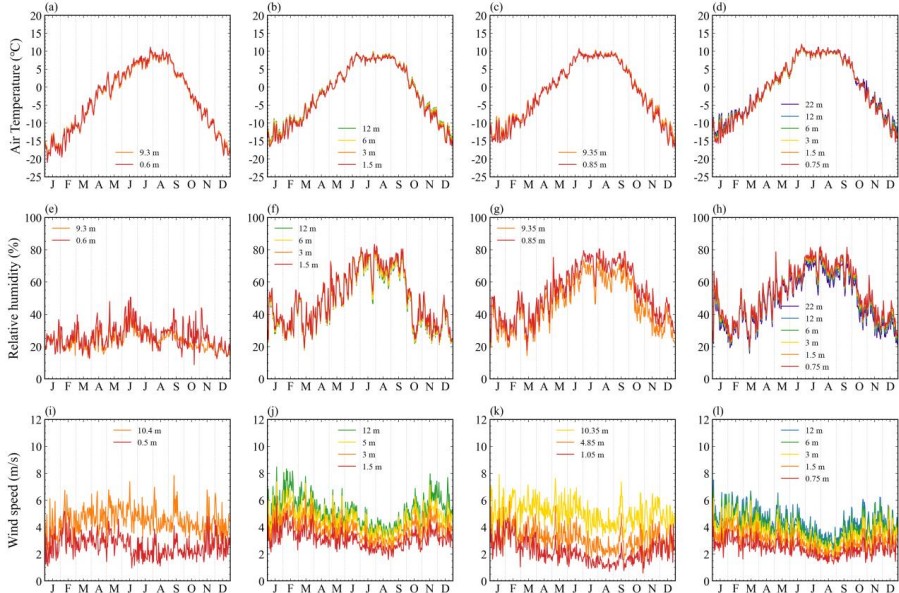

Figure 4. Daily mean of temperature at different heights of the PBL tower at NewD66 (a), Amdo (b), MS3478 (c) and BJ (d), with corresponding relative humidity (efgh) and wind speed (ijkl) at each station.

Figure 4 presents an annual cycle series, comprising daily mean values derived from gradient observations. At the NewD66 station, winter temperatures are demonstrably lower than at the other stations. In summer, the peak is reached at a later point in time and the duration is shorter. Other stations reach their highest temperatures in June and remain high until early September. With the onset of the monsoon season, continuous water vapour is transported from the ocean to the interior of the TP, resulting in a warming and humidification region. The stations begin to humidify rapidly in early May, reaching a peak in July with daily average relative humidity up to 80%. They then dry rapidly by the end of September, with the end of the monsoon season. The NewD66 station is a plateau bareland, and is only humid for a short period in June, the rest of the year the relative humidity is below 30%. Wind speeds at all stations show a pattern of being lower in summer and higher in winter. The TP is characterised by prevailing westerlies in winter, with daily mean wind speeds reaching 7 m/s. In contrast, the summer months are more humid with frequent changes in weather systems, resulting in average surface wind speeds of less than 4 m/s. Wind speeds at the NewD66 station are lower in winter than at other stations, but higher in summer.

Wind directions at the TP differ significantly between the monsoon and non-monsoon seasons (Fig. 5). Influenced by the South Asian and East Asian summer monsoons, northeast winds dominate during the monsoon season, accounting for 39% of the observations. In winter, there are fewer weather systems and most of the winds are steady westerly winds, with westerly winds accounting for up to 83% of the observations at different stations. MS3478 records different wind directions than the other stations. It has a consistently high frequency of southeast winds in both summer and winter, probably due to the surrounding topography.

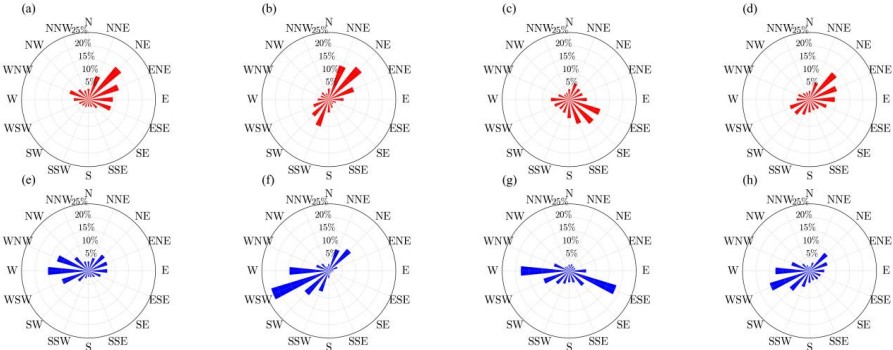


Figure 5. Wind direction during the monsoon season at NewD66 (a), Amdo (b), MS3478 (c), and BJ (d),
compared to their wind direction during the non-monsoon season (e-h).
**3.2 Radiation fluxes**
Downward shortwave radiation (Rsd) is the primary source of energy at the surface, and all
stations show similar diurnal patterns of Rsd. Observations indicate that the TP begins to absorb solar
radiation at 07:00 and stops completely at 21:00 as the sun sets (Fig. 6). The maximum Rsd occurs in
the afternoon and reaches 732 W/m². Seasonal variations demonstrate that Rsd is lowest in winter,
increases continuously until spring and peaks in May. This is followed by a decrease during the
monsoon season, which is likely due to the effects of frequent summer rainfall on the plateau and cloud
cover. Since there is little variation in Rsd, differences in upward shortwave radiation (Rsu) are mainly
due to variations in surface albedo. The trend in Rsu at all stations is consistent with that of Rsd, with
the highest values observed in May, a decrease during the summer months and the minimum in winter.
The start and end times of Rsu at each station align with those of Rsd, as illustrated in Figure 5c.
MS3478 and BJ stations exhibit slightly lower Rsu values. MS3478 is plateau wetland, and the area
near BJ station is a pasture with extensive vegetation cover in summer, which does not strongly reflect
sunlight. Consequently, the lower albedo at these two stations results in relatively lower Rsu values.
Longwave radiation is influenced by factors such as water vapour content, aerosol content, temperature
stratification and cloud thickness. It shows more pronounced seasonal variations than shortwave
radiation. The lowest levels of downward longwave radiation (Rld) are observed in winter, while the
highest levels are observed in summer. The diurnal variation of Rld is the smallest of the four radiation
components, fluctuating within a range of 45 W/m². Upward longwave radiation (Rlu) is minimal in
January, increases steadily until June, and stabilises at 370 W/m² during the summer months (June to
August). At station MS3478, Rlu is relatively lower, remaining around 350 W/m² during the summer.
The minimum of Rlu occurs at 08:00, coinciding with the lowest surface temperatures. The maximum
values occur in the afternoon and show greater diurnal variations compared to Rld, with fluctuations up
to 140 W/m². Net surface radiation exhibits a clear diurnal cycle, with peaks in the afternoon and
slightly negative values at night. This highlights the process of surface heating the atmosphere during
the day and cooling it at night. Annual variations show that net surface radiation is highest in summer
and lowest in winter.

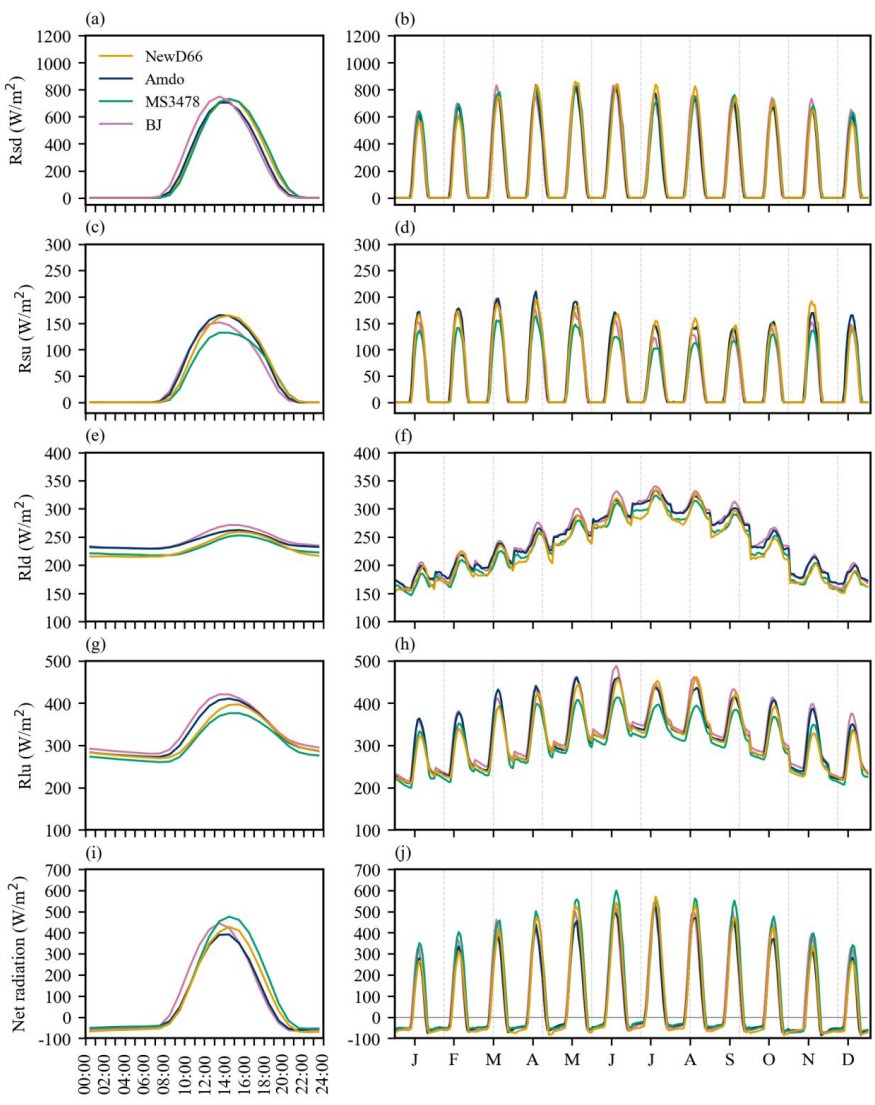

Figure 6. Diurnal variations (the first column) and monthly variations of the diurnal (the second column) downward shortwave radiation (a-b), upward shortwave radiation (c-d), downward longwave radiation (e-f), upward longwave radiation (g-h) ,and net radiation (i-j) at the 4 stations.

**3.3 Sensible heat and latent heat fluxes**

Sensible heat flux (SH) and latent heat flux (LH) are the primary forms of energy transfer between the surface and the atmosphere. Fig. 7 illustrates the monthly diurnal cycles and annual variations of SH and LH at each station. SH is influenced by the temperature difference between the surface and the atmosphere, as well as weather systems associated with precipitation, leading to numerous transient fluctuations. Seasonal variations demonstrate stronger SH at each station in spring and weaker in



summer. The daily peaks occur at 14:00, with negative values at night in winter, indicating clear
temperature inversions. While net radiation increases steadily in spring, the energy absorbed by the
surface before March is mainly used to melt snow. The onset of the monsoon season in May is
accompanied by a rapid increase in water vapour on the TP, with a gradual dominance of LH transfer.
Consequently, SH peaks in April at each station, with MS3478 (a plateau wetland) peaking in February.
The Asian monsoon system transports substantial moisture into the plateau during the summer,
resulting in low peaks in sensible heat flux at all stations, with a daily average of 20 W/m². In summer,
the NewD66 station experiences the highest SH, while the MS3478 station experiences the lowest SH,
reflecting the differences in water vapour content between the plateau bareland and plateau wetland.

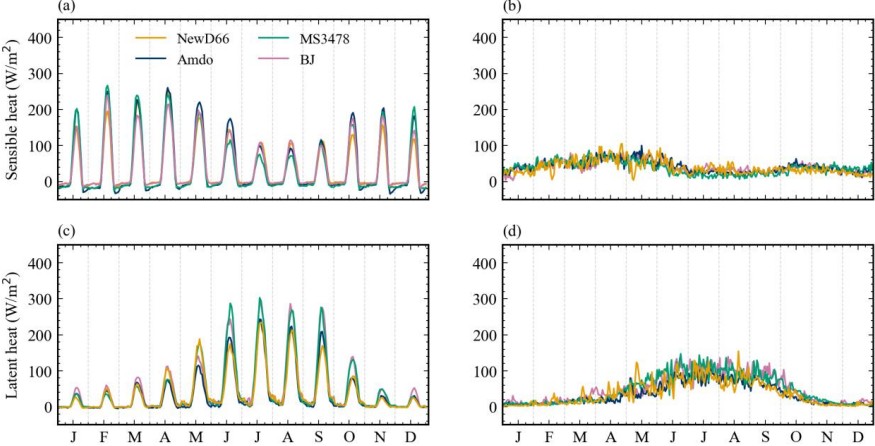

Figure 7. Monthly variations of the diurnal (the first column) and daily mean (the second column)
sensible heat (a-b), latent heat (c-d), and total heat flux (e-f) at the 4 stations.

The seasonal variation trend of LH is in opposition to that of SH. During the winter, the plateau is
characterised by a relatively cold and dry climate, with LH peaks not exceeding 60 W/m² at all stations.
As the monsoon season commences in May, LH rapidly rises, maintaining levels above 100 W/m²
throughout the summer until it begins to decline in October. Peaks in LH occur in July at all stations,
with MS3478's peak appearing earlier in June due to its higher moisture content and remaining high
throughout the summer. In winter, nighttime LH approaches zero, but retains weak positive values in
summer, indicating that no clear temperature inversions occur at night, and even contributing to heating
the atmosphere in the early hours of the night. Fig. 7d shows significant fluctuations in LH during the
summer months, which are attributed to the frequent alterations in weather patterns. The diurnal cycle
of LH is also influenced by the occurrence of extreme rainfall episodes.
**3.4 Soil hydrothermal characteristics**

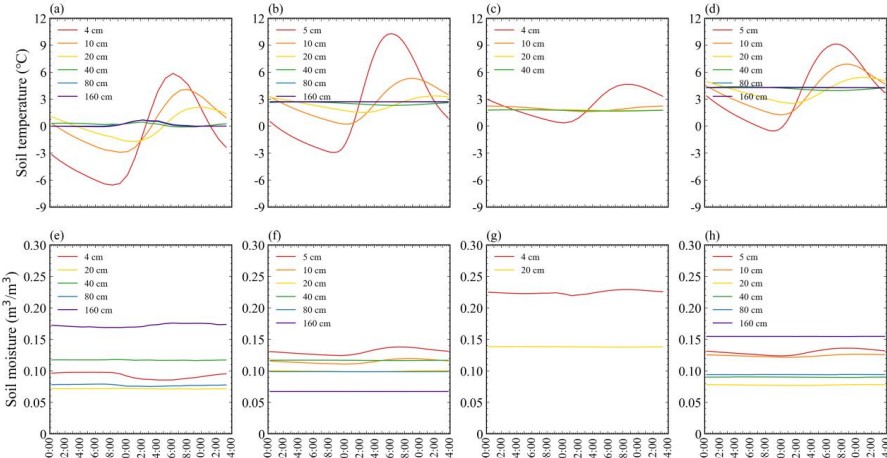

Figure 8. Layered soil temperature at NewD66 (a), Amdo (b), MS3478 (c), and BJ (d) observation, and
corresponding layered soil moisture (efgh) at each station.

Soil thermal and moisture variations have a significant impact on energy and moisture exchange between the surface and the atmosphere. Due to varying observation conditions, soil observation depths differ between stations. Fig. 8 shows that all stations exhibit a diurnal cycle in soil temperature, with cooling at night and warming during the day. Soil temperature variation diminishes with increasing depth, with 4cm, 10cm, and 20cm depths showing a 2-hour delay in heating and cooling cycles. Significant spatial variation is observed due to differences in soil type, porosity, and substrate. The soil temperature at NewD66 station tends to be lower than at other sites, with 4-cm depths reaching a minimum of -6.56 °C at night. This is due to a lack of moisture in the surrounding air, which results in a greater daily heating effect, reaching 5.86 °C by 16:00. At 160 cm, a slight midday fluctuation is observed, which is not seen at other stations. MS3478, a plateau wetland, exhibits a diurnal variation of 4.29 °C at 4-cm depth, with a slight rise at 10 cm at 22:00, and minimal diurnal variation below 10 cm.

The diurnal cycle of soil moisture is weak at the shallowest layer, with fluctuations not exceeding 0.015 m³/m³. Below 10 cm, soil moisture shows no clear diurnal cycle. The overall soil moisture at NewD66 station is relatively low, but it does not vary linearly with depth. For example, 5-cm depth soil moisture is 0.09 m³/m³, while 160-cm depth moisture remains consistently at 0.17 m³/m³. The soil moisture data from the BJ station exhibits a comparable pattern, with slight fluctuations at the 5-cm depth, maintaining an average of approximately 0.13 m³/m³, and negligible variation at the 160-cm depth, remaining at 0.15 m³/m³. In contrast, the soil moisture at the 160-cm depth from Amdo is considerably lower than that observed in the shallow layers, likely due to the presence of a permafrost layer, which results in a consistently low liquid water content. MS3478's soil moisture exhibits no significant diurnal variation, with 4-cm depth soil moisture averaging 0.22 m³/m³ and 20-cm depth soil moisture averaging 0.13 m³/m³, which are both significantly higher than those observed at other stations.

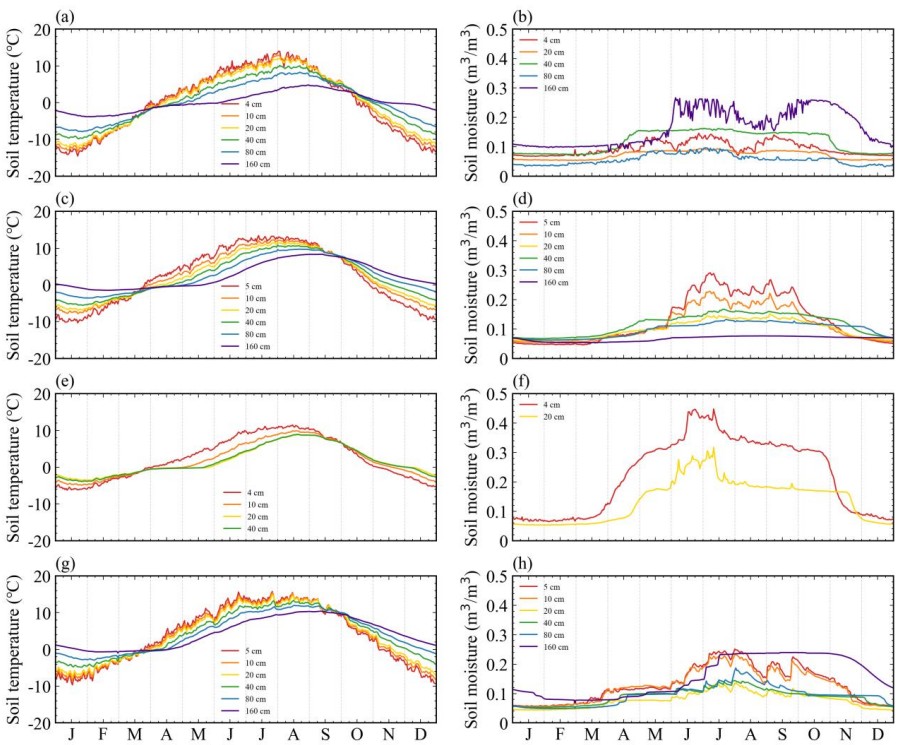

Figure 9. Daily mean soil temperature (a) and soil moisture (b) at NewD66,
same in Amdo (cd), MS3478 (ef), and BJ (gh).

Soil temperature and moisture exhibit clear annual variations at each station, with soil temperature delaying more with increasing observation depth (Fig. 9). Surface soil temperatures peak in July, 40-cm depth temperatures in August, and 160-cm depth temperatures in September. The greatest annual variation in soil temperature is observed in NewD66's 4-cm depth soil temperature, reaching 28.56 ℃, while the smallest is observed in MS3478's, at 17.74 ℃. A similar trend is observed in soil moisture, with higher levels observed in summer and lower levels in winter. Soil moisture increases in spring due to the melting of snow and ice, and in summer due to precipitation. In the absence of rainfall, surface evaporation and runoff results in a rapid decrease in soil moisture. There are significant differences in soil moisture between stations, with substantial fluctuations in surface moisture, indicating vertical variation in the soil moisture profile. NewD66's 4-cm depth soil moisture exhibits minimal annual variation, increasing by only 0.05 m³/m³ in summer. However, 160-cm depth moisture rapidly increases to 0.24 m³/m³ in summer, then gradually decreases in November, coinciding with the point at which 160-cm depth soil temperatures exceed 0 ℃. The soil moisture at all layers of Amdo is below 0.1 m³/m³ in winter, with 5-cm depth moisture reaching 0.29 m³/m³ in summer, and decreasing with increasing depth. MS3478's humid air contributes to higher soil moisture, with 4-cm depth moisture maintaining levels above 0.3 m³/m³ from May to October, even exceeding 0.8 m³/m³ at times. BJ's 4-cm depth moisture rises in April, peaking in August at 0.25 m³/m³. The lowest 20-cm depth moisture is observed year-round, at only 0.12 m³/m³ in July. The 160-cm depth moisture rapidly rises in June, maintaining 0.24 m³/m³ until November.




**4. Data availability**

Near-surface micrometeorology, soil hydrothermal characteristics, and turbulent flux observation
datasets are openly available at https://doi.org/10.11888/Meteoro.tpdc.270010 (Hu et al., 2019) and
https://doi.org/10.11888/Atmos.tpdc.300325 (Wang et al., 2023). Following a rigorous screening
process that included the identification and elimination of missing values, the physical range check,
and the manual quality control, valid data in CSV format is available for download. The file names
adhere to the convention of station_datatype_year.csv, and all data files utilise UTC+8 as the
designated time standard. Each CSV-formatted data file name includes the name of the variable, the
observed height/depth, and the unit.

**5. Conclusion and Discussion**

As demonstrated in previous sections, the NPCE reliably reflects the characteristics of various
variables across different underlying surface and temporal scales, thereby showcasing the reliability
and completeness of the data. The challenging environment and dispersed stations on the CTP make it
difficult to maintain equipment, which makes the data collected valuable. The scarcity of field
observation data has long been a barrier to in-depth research on the TP. Thus, long-term observations
and scientific data sharing are crucial for research on the plateau. This paper introduces land-
atmosphere interaction data from several observation stations centred around Nagqu. The dataset
comprises 9 years (2014-2022) of hourly continuous series of near-surface meteorological, radiation,
turbulent flux, and soil hydrothermal characteristics. This enables effective exploration of the entire
process of land-atmosphere interactions on the plateau. Additionally, all data were manually checked
for outliers and subjected to quality control. The long-term half-hourly dataset from NPCE provides the
opportunity to conduct high-precision numerical simulations at the local level. This enables in-depth
research to be conducted on the mechanisms by which the land and atmosphere interact on the plateau
and also provides support for comprehensive research into the plateau's energy and water cycles and
climate change.
The NPCE is a comprehensive research platform with the primary goal of observing and studying
the impacts of weather and climate change on the Tibetan Plateau, water resource utilisation, ecological
environmental protection, and human activities. The main objective of the NPCE is based on a plateau
climate and the water-energy cycle. In recent years, a range of processes have been the focus of
detailed analysis, including non-homogeneous terrain turbulence transfer mechanisms, radiation
balance components and energy allocation processes. The analysis has been informed by long-term in
situ meteorological observations (Huang et al., 2016). The spatial and temporal variability of SH and
LH under complex topographic forcing is a particular highlight (Xie et al., 2018), and energy closure in
the central part of the northern Tibetan plateau is further discussed (Li et al., 2015; Li et al., 2016).
Satellite-derived products have been utilised to address the spatial representation limitations inherent in
field observations (Ma et al., 2014; Ma et al., 2009; Ma et al., 2006). Furthermore, ground-based
observations were utilised to validate and calibrate the remote sensing estimates of sensed LH, thereby
improving the accuracy of the remote sensing algorithms (Zhong et al., 2019). A distributed sensor
network was utilised to monitor soil temperature and humidity gradients, thereby enabling the
systematic quantification of the disparities in soil moisture and heat fluxes between permafrost and
seasonal permafrost. In these areas, phase change LH associated with freeze-thaw transitions have been
demonstrated to influence surface energy distribution (Liu et al., 2015; Fu et al., 2022). A



comprehensive analysis of the pattern of boundary layer changes on the CTP has been conducted
utilising PBL towers, as well as MWR and radiosonde data (Wang et al., 2025; Sun et al., 2021;
Sun et al., 2020). The integration of multi-source observations (Near-surface meteorological data, EC
system, boundary layer observation, and satellite inversion) has modified the model representation of
land-atmosphere energy balance processes, and further refinement of the parameterisation scheme has
been applied to soil freeze-thaw processes (Deng et al., 2021). The enhanced numerical simulation
accuracy of blowing snow processes in the CLM model facilitates a more profound comprehension of
surface processes and cryosphere changes in local area (Xie et al., 2023; Xie et al., 2021).
The data presented in the paper is the average result after filtering out missing values. However, in
practical applications, considering the difficulty of instrument maintenance, there may inevitably be
anomalies or data discontinuities. As a key observation project on the CTP, the NPCE is implementing
an ambitious plan to enhance its monitoring capabilities for understanding global climate dynamics due
to its high altitude and ecological sensitivity. This initiative focuses on acquiring advanced
instrumentation and expanding monitoring sites to deepen insights into atmospheric, hydrological, and
ecological processes, particularly in the CTP and the upper Nu River (Salween River) basin. For
instrument acquisition, the station will deploy Raman lidar systems, which are highly specialized for
simultaneous detection of water vapor and aerosols in the lower atmosphere. These Raman lidar
systems utilize laser-based spectroscopy to provide high-resolution measurements of water vapor
concentrations, enabling detailed analysis of moisture transport and its influence on regional climate
patterns. Additionally, the same Raman lidar systems will measure aerosol properties, such as particle
size, concentration, and distribution, which are essential for studying air quality, aerosol-cloud
interactions, and their impact on radiative forcing in the plateau's unique environment. To complement
these atmospheric measurements, the station will acquire advanced greenhouse gas analyzers
specifically designed to monitor carbon dioxide and methane fluxes from soil and vegetation
respiration, providing critical data on biogeochemical cycles and the carbon dynamics of high-altitude
ecosystems under changing climatic conditions.
The site expansion plan involves establishing new monitoring stations on the CTP and the upper Nu
River basin to broaden the scope of environmental observations. The CTP, with its expansive high-
altitude grasslands, offers a unique setting to study the interplay between atmospheric processes and
ecosystem responses to climate change. Similarly, the upper Nu River basin, a vital hydrological region,
will host new sites to investigate water cycles, soil moisture dynamics, and vegetation responses to
environmental shifts. By expanding coverage in the upstream Nu River, the station aims to capture
spatial and temporal variations in hydrological and ecological parameters, supporting research on water
resource management, ecosystem resilience, and climate adaptation strategies in this ecologically
sensitive area.

**Acknowledgements**
This work was jointly supported by Science and Technology Projects of Xizang Autonomous
Region, China (XZ202501JD0022), the National Natural Science Foundation of China (42330609,
U2442207), the Youth Innovation Promotion Association of Chinese Academy of Sciences (2021427),
the West Light Foundation (xbzg-zdsys-202409) of the Chinese Academy of Sciences, Key talent
project in Gansu and Central Guidance Fund for Local Science and Technology Development Projects
in Gansu (No.24ZYQA031).



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
