# Peer review of "Long-term Land-Atmosphere Energy and Water Exchange Observational Dataset over central Tibetan Plateau"

_Earth System Science Data, 2025_

## Referee Comment (RC1)

**Review of Manuscript ESSD-2025-356 by Yu et al., 2025 "Long-term Land–Atmosphere Energy and Water Exchange Observational Dataset over central Tibetan Plateau"**

General comments

This manuscript presents a comprehensive, high-temporal-resolution (hourly) observational dataset from the Nagqu Plateau Climate and Environment Observation and Research Station (NPCE) on the central Tibetan Plateau (CTP), covering the period 2014–2022. The dataset includes near-surface meteorology, radiation, turbulent fluxes, and soil hydrothermal measurements from four stations representing different land cover types. The manuscript is well-structured, clearly written, and provides sufficient methodological and contextual detail to support the publication of the dataset.

Specific comments:

1. Appropriateness for Supporting Data Publication

The manuscript effectively serves as a companion paper to the dataset. It includes:

- A clear introduction justifying the need for such data in a poorly observed region.

- Detailed descriptions of station locations, instrumentation, data processing, and quality control procedures.

- Visualizations (figures) illustrating diurnal and seasonal variations in key variables, which help users understand the data characteristics.

- References to existing data repositories (DOI links) where the data are hosted.

The paper aligns well with ESSD's goal of documenting and promoting the reuse of high-value scientific datasets.

2. Significance: Uniqueness, Usefulness, and Completeness

Highly significant.

- Uniqueness: The CTP is a critical yet underobserved region due to its extreme altitude and harsh conditions. This dataset provides long-term, high-resolution observations from multiple sites with varied land cover—a rare and valuable resource.

- Usefulness: The data are directly relevant to:

  - Model validation and improvement (e.g., land-surface, regional climate, and hydrological models).

- Studies on land–atmosphere interactions, energy/water cycles, climate change impacts, and monsoon dynamics.

  - Satellite product validation and remote sensing algorithm development.

- Completeness: The dataset spans 9 years with hourly resolution, covers multiple variables, and includes quality flags. Data gaps and quality control steps are transparently documented.

**3. Data Quality**

High quality.

- The authors describe a multi-step quality control process:

  - Range checks based on Tibetan Plateau climate conditions.

  - Manual inspection to remove outliers or erroneous intervals.

  - Use of standard missing value markers (NaN, -9999).

  - Quality classification for turbulent fluxes (levels 0–2).

- Instrumentation is well-documented (Table 1), and consistent sensor types are used across stations to ensure comparability.

- Figures 2–9 demonstrate plausible diurnal and seasonal cycles, consistent with known Tibetan Plateau climatology (e.g., strong solar radiation, monsoon influence, soil freeze–thaw dynamics).

**4. Publication Quality**

Very good, with minor suggestions for improvement.

- The manuscript is well-organized, with clear sections and informative figures.

- Data are openly available via two DOIs, and file naming conventions and time standards are clearly explained.

- The discussion and conclusion sections contextualize the data within broader research efforts and future plans (e.g., expanded instrumentation and sites).

Minor Suggestions:

- Consider including a summary table of data availability per station and variable (e.g., percentage of valid data per year) to help users assess data continuity.

- Clarify whether the wind direction data (Fig. 5) are from a specific height or an average across levels.

- Standardize station naming in figures and text (e.g., "NewD66" vs. "Kekexili").

- Briefly mention any known limitations (e.g., sensor drift, maintenance challenges) beyond what is already described.

- Comment on why soil heat flux is not reported, which is critical flux to close surface energy balance. (I understood soil heat flux is a standard observation variable in such important climate stations).

- Comment on why $CO_2$ flux is not reported, for the same reason as for soil heat flux.

- Fig. 7. The total heat flux (e-f) plots are missing.

- For soil hydrothermal measurements, some comments related to in-situ observation of soil texture, soil hydraulic and thermal properties would be highly valuable for data applications. If the authors have not done such analysis, they may refer to relevant studies that have done so at some locations near the mentioned stations.

**Overall Recommendation**

Accept with minor revisions.

This dataset is trustworthy and highly useful for the research community. It fills a critical gap in high-altitude land–atmosphere interaction data and is backed by rigorous quality control and clear documentation. The manuscript effectively supports the data publication and encourages reuse across multiple disciplines.

*(I suggest that the editorial office check, before the acceptance of the manuscript, the accessibility and completeness of the uploaded datasets for which I have little time.)*

---

## Author Comment (AC1)

Reviewer 1

General comments

This manuscript presents a comprehensive, high-temporal-resolution (hourly) observational dataset from the Nagqu Plateau Climate and Environment Observation and Research Station (NPCE) on the central Tibetan Plateau (CTP), covering the period 2014–2022. The dataset includes near-surface meteorology, radiation, turbulent fluxes, and soil hydrothermal measurements from four stations representing different land cover types. The manuscript is well-structured, clearly written, and provides sufficient methodological and contextual detail to support the publication of the dataset.

Specific comments:

1. Appropriateness for Supporting Data Publication

The manuscript effectively serves as a companion paper to the dataset. It includes:

- A clear introduction justifying the need for such data in a poorly observed region.

- Detailed descriptions of station locations, instrumentation, data processing, and quality control procedures.

- Visualizations (figures) illustrating diurnal and seasonal variations in key variables, which help users understand the data characteristics.

- References to existing data repositories (DOI links) where the data are hosted. The paper aligns well with ESSD's goal of documenting and promoting the reuse of highvalue scientific datasets.

2. Significance: Uniqueness, Usefulness, and Completeness Highly significant.

- Uniqueness: The CTP is a critical yet underobserved region due to its extreme altitude and harsh conditions. This dataset provides long-term, high-resolution observations from multiple sites with varied land cover—a rare and valuable resource.

- Usefulness: The data are directly relevant to:

 - Model validation and improvement (e.g., land-surface, regional climate, and hydrological models).

 - Studies on land–atmosphere interactions, energy/water cycles, climate change impacts, and monsoon dynamics.

 - Satellite product validation and remote sensing algorithm development.

- Completeness: The dataset spans 9 years with hourly resolution, covers multiple variables, and includes quality flags. Data gaps and quality control steps are transparently documented.

3. Data Quality

High quality.

- The authors describe a multi-step quality control process:

 - Range checks based on Tibetan Plateau climate conditions.

 - Manual inspection to remove outliers or erroneous intervals.

 - Use of standard missing value markers (NaN, -9999).

 - Quality classification for turbulent fluxes (levels 0–2).

- Instrumentation is well-documented (Table 1), and consistent sensor types are used across stations to ensure comparability.

- Figures 2–9 demonstrate plausible diurnal and seasonal cycles, consistent with known Tibetan Plateau climatology (e.g., strong solar radiation, monsoon influence, soil freeze–thaw dynamics).

4. Publication Quality

Very good, with minor suggestions for improvement.

- The manuscript is well-organized, with clear sections and informative figures.

- Data are openly available via two DOIs, and file naming conventions and time standards are clearly explained.

- The discussion and conclusion sections contextualize the data within broader research efforts and future plans (e.g., expanded instrumentation and sites).

Reply: Thank you for your recognition of this study. We will continue to overcome difficulties and challenges to ensure that the data quality from the Nagqu observation network meets the standards required for scientific research. This dataset not only serves as a scientific resource but also represents a successful effort to extend the Tibetan Plateau observation network toward the central plateau.

Minor Suggestions:

1.Consider including a summary table of data availability per station and variable (e.g., percentage of valid data per year) to help users assess data continuity.

Reply: The Nagqu observation network undergoes routine maintenance every two months. With the exception of a few stations and specific variables, the network maintains data availability exceeding 95% for most observations, including conventional surface measurements. The availability of turbulence-related fluxes and soil temperature and moisture observations remains above 85%. It is notable that at the NewD66 station, only about 70% of the data on sensible heat flux, latent heat flux, and soil moisture meet quality standards. Data with availability below 50% (such as soil heat flux at station MS3478) are excluded from presentation in this study. To facilitate user evaluation of data quality, entries with availability below 70% are marked with the mark "*" in Table 1, and detailed explanations are provided beneath the table in Line 199: "All data presented herein exhibit availability greater than 50%, with the mark "*" specifically indicating availability below 80%."

2.Clarify whether the wind direction data (Fig. 5) are from a specific height or an average across levels.

Reply: Thank you for your suggestion. We have now incorporated the wind direction heights from the four observation stations into the caption of Figure 5 at line 272: "Figure 5. Wind direction during the monsoon season at NewD66 (a, 10.4 m), Amdo (b, 12.0 m), MS3478 (c, 10.35 m), and BJ (d, 12.0 m), compared to their wind direction during the non-monsoon season (e-h)."

3.Standardize station naming in figures and text (e.g., "NewD66" vs. "Kekexili").

Reply: Thank you for your suggestion. In this article, "NewD66" refers to the name of the observation station, while "Kekexili" is a geographical term that specifies the location of the station, only the station names are used throughout the text. To avoid potential ambiguity, the corresponding region of each station has been explicitly listed under the respective station name in Table 1.

4.Briefly mention any known limitations (e.g., sensor drift, maintenance challenges) beyond what is already described.

Reply: Thank you for your valuable feedback. We have expanded the discussion on factors influencing data accuracy in Line 429: "The challenging environment and dispersed stations on

the CTP make it difficult to maintain equipment. Frequent extreme weather events contribute to recurrent instrument malfunctions, such as sensor drift, calibration error, environmental interference, and power supply issues."

5.Comment on why soil heat flux is not reported, which is critical flux to close surface energy balance. (I understood soil heat flux is a standard observation variable in such important climate stations).

Reply: Thank you for your valuable suggestions. As soil heat flux is a key physical quantity in the calculation of surface energy balance, we have incorporated additional data visualizations related to it in Fig. 7 ef, with a detailed description provided in Line 337: "SHF is defined as the heat transfer from the land surface to deeper soil layers, is a key component of the surface energy balance. As illustrated in Figure 7ef, the seasonal variation in SHF is relatively moderate compared to SH and LH, with seasonal maxima remaining below 100 W/m². A clear diurnal cycle is observed, characterized by a peak in the afternoon and a minimum during the night. The largest diurnal variation in SHF occurred at Amdo Station from January to April. In contrast, during May to September when surface thermal activity is most intense, Newd66 Station consistently exhibited high levels of diurnal variability. In terms of daily mean values, SHF was highest from May to July and lowest in December. It is noteworthy that Newd66 Station showed the most pronounced diurnal fluctuation, which may be attributed to its plateau bareland underlying surface, which is characterised by a simple composition and high sensitivity to external disturbances. Furthermore, interannual variability was greatest at Newd66 Station, where the amplitude of SHF reached 40 W/m², while at BJ Station, the variation was comparatively subdued, with an amplitude of only 22.7 W/m² throughout the annual cycle."

6.Comment on why CO2 flux is not reported, for the same reason as for soil heat flux.

Reply: Thank you for your suggestion. We have incorporated the $CO_2$ variable into the dataset, with the corresponding data description provided in Table 2. Relevant figures and descriptions have been added in Line 350:

[Figure]

Figure 8. Monthly variations of the diurnal (a) and daily mean (b) $CO_2$ concentration at the 4 stations.

As shown in Figure 8a, $CO_2$ concentrations exhibit a diurnal cycle characterized by an increase during nighttime and a decrease during daytime. This cycle shows minimal monthly variation across the observation period. The NewD66 station displays only a weak diurnal signal between June and August, which may be attributed to the sparse vegetation over the bareland underlying surface, resulting in a relatively limited influence on local $CO_2$. In contrast, the BJ station, situated near pastureland and remote from human activities, shows relatively lower $CO_2$

concentrations compared to the other stations, all of which are located within a few kilometres of the Qinghai–Tibet Highway. Figure 8b illustrates the annual variation in $CO_2$ concentration. The BJ station consistently exhibits lower $CO_2$ levels throughout the year, with more pronounced fluctuations during summer. Seasonal amplitude is greater at Amdo in summer and at MS3478 in winter. The most substantial annual variation is observed at NewD66, where $CO_2$ concentrations reach a minimum in April and peak in June.

7.Fig. 7. The total heat flux (e-f) plots are missing.

Reply: Thank you for your reminder. We have replaced the content in Fig. 7 ef with soil heat flux data to ensure a comprehensive representation of the relevant variables in the surface energy balance.

8.For soil hydrothermal measurements, some comments related to in-situ observation of soil texture, soil hydraulic and thermal properties would be highly valuable for data applications. If the authors have not done such analysis, they may refer to relevant studies that have done so at some locations near the mentioned stations.

Reply: Thank you for your valuable suggestions. We have added an analysis of different underlying surface types at Line 139 to provide readers with a more comprehensive understanding of the fundamental characteristics of each station: "The locations in Figure 1 are BJ, Amdo, NewD66, and MS3478. These represent different substrate characteristics, namely plateau grassland, plateau meadow, plateau wetland, and plateau bareland. Since plant roots are primarily concentrated in the shallow soil layers of the plateau, root biomass in the 0–10 cm layer accounts for 70% of the total root biomass (Su et al., 2021). This leads to pronounced seasonal variations in the surface soil at each observation site. In addition to vegetation, variations in soil temperature and moisture are influenced by precipitation, soil properties, and underlying permafrost. The correlation between soil temperature and air temperature gradually decreases with increasing depth (Wang et al., 2023). Consequently, the different underlying surfaces within the Nagqu observation network represent distinct environmental features of the TP, their interannual variation trends and influencing factors exhibit notable differences (Sun et al., 2015)."

Overall Recommendation

Accept with minor revisions. This dataset is trustworthy and highly useful for the research community. It fills a critical gap in high-altitude land–atmosphere interaction data and is backed by rigorous quality control and clear documentation. The manuscript effectively supports the data publication and encourages reuse across multiple disciplines. (I suggest that the editorial office check, before the acceptance of the manuscript, the accessibility and completeness of the uploaded datasets for which I have little time.)

Reviewer 2

The Tibetan Plateau holds significant importance in East Asian and global climate researches, yet its complex terrain and poor observational conditions have constrained related studies. In this paper, the authors release multi-year meteorological and soil observation data from the Nagqu region, filling a gap in observations across the central plateau and providing robust data support for climate research on the Tibetan Plateau. The article details rigorous quality control and validation procedures for the dataset, demonstrating its high quality. Overall, I believe this manuscript meets publication standards and recommend acceptance in its current version.

Reply: Thank you for your positive feedback on our work. We will continue to develop the Nagqu observation network, with ongoing efforts to expand the range of data types and improve data availability. The progressive enhancement of the observation system over the Tibetan Plateau will substantially advance scientific research on land–atmosphere interactions in this critical region.

---

## Author Response (AR2)

Public justification (visible to the public if the article is accepted and published):
Thank you for the upload of the revised manuscript that addresses the reviewers' comments. Based on the revised manuscript, I have two comments that should be addressed.

Specific comment: Figure 8 shows CO2 concentrations but not fluxes from the IRGA. Based on the mean concentrations it appears that these differences might be mainly due to calibration offsets. It appears unlikely that two stations have 100% difference in CO2 concentrations (this should be checked by converting to a more standard/ comparable unit like ppm, given that temperature and pressure also factor into these differences. This would also affect any annual changes, whereas PPM would be indifferent to annual temperature change). I understand that calibration in these locations can be difficult, but these CO2 differences should be discussed in the manuscript (especially with respect on what this might mean for fluxes in the dataset).

Reply: Thank you for the valuable suggestion. As the BJ station initiated the earliest eddy covariance observations in the network, its prolonged operation led to a calibration delay that introduced measurable errors in the $CO_2$ concentration data. This issue was addressed through a recalibration performed in May 2022. To uphold the data integrity of the Nagqu observation network, we have made the $CO_2$ data from the BJ station for the period 2022–2025 publicly available. New Fig.8 presenting the data after quality control restoration is provided in Row 345, with the units consistently converted to ppm as requested.

Technical Comment: Replace CO2/H2O analysis meter with "CO2/ H2O Gas Analyzer" in Table 1

Reply: In response to your comment, the term "$CO_2$/$H_2O$ analysis meter" in Table 1 has been revised to "$CO_2$/$H_2O$ Gas Analyzer".